# A case of T-cell acute lymphoblastic leukemia in retroviral gene therapy for ADA-SCID

Daniela Cesana [1,22], Maria Pia Cicalese[1,2,3,22], Andrea Calabria [1], Pietro Merli[4], Roberta Caruso[4], Monica Volpin [1], Laura Rudilosso[1], Maddalena Migliavacca[1,2], Federica Barzaghi[1,2], Claudia Fossati[1], Francesco Gazzo[1], Simone Pizzi[5], Andrea Ciolfi [5], Alessandro Bruselles[6], Francesca Tucci[1,2], Giulio Spinozzi [1], Giulia Pais [1], Fabrizio Benedicenti[1], Matteo Barcella[1,7], Ivan Merelli[1,7], Pierangela Gallina[1], Stefania Giannelli[1], Francesca Dionisio[1], Serena Scala[1], Miriam Casiraghi[1], Luisa Strocchio[4], Luciana Vinti[4], Lucia Pacillo [8], Eleonora Draghi[9], Marcella Cesana[10,11], Sara Riccardo[10,12], Chiara Colantuono [10,12], Emmanuelle Six [13], Marina Cavazzana[13], Filippo Carlucci[14], Manfred Schmidt[15,23], Caterina Cancrini[8,16], Fabio Ciceri [1,3,17], Luca Vago [3,9,17], Davide Cacchiarelli [10,18,19], Bernhard Gentner [1,17], Luigi Naldini [1,3], Marco Tartaglia[5], Eugenio Montini [1,24], Franco Locatelli [20,21,24] & Alessandro Aiuti [1,2,3,24] ✉

Hematopoietic stem cell gene therapy (GT) using a γ-retroviral vector (γ-RV) is an effective treatment for Severe Combined Immunodeficiency due to Adenosine Deaminase deficiency. Here, we describe a case of GT-related T-cell acute lymphoblastic leukemia (T-ALL) that developed 4.7 years after treatment. The patient underwent chemotherapy and haploidentical transplantation and is currently in remission. Blast cells contain a single vector insertion activating the LIM-only protein 2 (*LMO2*) proto-oncogene, confirmed by physical interaction, and low Adenosine Deaminase (ADA) activity resulting from methylation of viral promoter. The insertion is detected years before T-ALL in multiple lineages, suggesting that further hits occurred in a thymic progenitor. Blast cells contain known and novel somatic mutations as well as germline mutations which may have contributed to transformation. Before T-ALL onset, the insertion profile is similar to those of other ADA-deficient patients. The limited incidence of vector-related adverse events in ADA-deficiency compared to other γ-RV GT trials could be explained by differences in transgenes, background disease and patient's specific factors.

Retroviruses are widely used as vectors for gene therapy (GT) approaches in humans. Gene therapy with autologous transduced hematopoietic stem/progenitor cell (HSPC-GT) has emerged as an effective treatment for several inherited diseases, including inborn errors of immunity (IEI)[1–5].

A known risk of HSPC-GT with γ-retroviral vector (γ-RV) is insertional mutagenesis caused by integration of vector enhancer sequences contained within the long-terminal repeats (LTR) that can induce activation of neighboring genes[6]. Such oncogenic events have been reported in SCID-X1, X-linked chronic granulomatous disease (X-CGD),

or Wiskott-Aldrich Syndrome (WAS) patients treated with γ-RV[7–14]. The oncogenic events occurred from 1.3 to 14.8 years after GT and were associated predominantly with insertional activation of the proto-oncogenes LIM domain only 2 (*LMO2*) for T-cell acute lymphoid leukemia or *MECOM* for myelodysplasia/myeloid leukemia[7–14].

Adenosine Deaminase-deficient Severe Combined Immunodeficiency (ADA-SCID) is a purine metabolic disease-causing impaired lymphocyte differentiation and function, which leads to severe, recurrent, opportunistic infections. HSPC-GT has been proven a safe and effective treatment in patients lacking a human leukocyte antigen (HLA)-matched related HSPC donor. Enzyme replacement therapy can be administered to ADA-SCID patients as a bridging treatment towards definite therapy whereas allogeneic hematopoietic stem cell transplantation (HSCT) from donors other than an HLA-matched subject is associated with higher morbidity and mortality[15,16]. Since 2000, 75 ADA-SCID patients have been reported to be treated worldwide with autologous CD34+ cells engineered ex vivo with γ-RV encoding ADA, usually following low-dose busulfan preparative regimen[2]. Survival today is 100% in line with metabolic correction, progressive immunological improvement and clinical amelioration. Treatment with autologous bone marrow (BM)-derived CD34+ cells transduced with γ-RV (Strimvelis) was approved in the EU in 2016[17] and long-term follow-up confirmed the persistence of engraftment and clinical benefit[18]. A relatively increased proportion of clones carrying vector insertion sites (IS) near *LMO2* and *MECOM* has been documented in the patients' reconstituted hematopoiesis, which remained stable and did not reach dominance or induce dysplasia[19–21]. However, we recently reported a single case of T-cell acute lymphoblastic leukemia related to GT in the analysis on long-term safety and efficacy of 43 ADA-SCID patients who received retroviral ex vivo HSPC GT[18]. Here we provide the detailed characterization of the leukemia features and investigate the potential factors that contributed to the event.

## Results

### Clinical findings

P21 was a male of European descent without a documented family history of malignancy, who presented with a CMV primary infection at the age of 4 months and was diagnosed with ADA-SCID for profound lymphopenia (lymphocytes 0.3 ×10^9/L) and biallelic pathogenic variants (c.455T>C [p.Leu152Pro] and c.478+6T>C) in the *ADA* gene (Table 1). He lacked a human leukocyte antigen (HLA)-identical sibling donor and at the age of 1 year was treated with autologous CD34+ cells transduced with γ-RV under Named Patient Program after low-dose busulfan conditioning[22] (Fig. 1). The patient received a medicinal product within target cell dose and transduction level (VCN 1.8 copies/genome), resulting in adequate hematopoietic engraftment (Table 1). P21's post-infusion course and hospitalization were uneventful. The only serious adverse event (SAE) was a symptomatic measles occurring 17 months after GT, which eventually led to the need for patient's hospitalization for fever.

Engraftment of gene-corrected cells, lymphocyte and CD3+ T cell counts as well as ADA enzymatic activity and metabolic detoxification (dAXP RBC, data not shown) were comparable to those of other patients along the follow-up (Supplementary Fig. 1). T-cell proliferation was normal after GT, immunoglobulins supplementation was discontinued 1 year after GT, with valid humoral response to most vaccines achieved (Table 1).

Routine follow-up (FU) evaluation until 4.5 years post-GT did not reveal laboratory or clinical abnormalities. However, at the age of 5.75 years (4.7 years after GT), the patient developed a T-cell acute lymphoblastic leukemia (T-ALL) (Fig. 1). Clinical onset was characterized by bruising and weakness. Blood tests showed WBC 291.3 × 10^9/L with immature lymphocytes (93%) and cytofluorimetric markers of T-ALL (Table 1). There was no central nervous system or mediastinal involvement. The blast population was transduced (1.18

vector copies/genome) (Table 1) and expressed low ADA activity (Supplementary Fig. 1c). The patient was enrolled in the AIEOP-BFM ALL 2017 protocol, receiving chemotherapy according to the high-risk protocol and was given allogeneic HSCT because of persisting high levels of minimal residual disease (MRD) at day 78. In the absence of a matched unrelated donor, the patient received a TCRαβ/CD19 depleted HLA-haploidentical HSCT from the mother. Donor engraftment was achieved promptly, and one donor lymphocyte infusion was administered at day +37 due to MRD persistence. Full negativity of MRD, 100% donor chimerism, in the absence of graft-versus-host disease, were achieved from month 2 to month 29. The patient is alive and well at the latest follow-up (month 33 post-HSCT as of February 12th, 2024) (Table 1).

### Molecular analyses and dynamics of T-ALL

Vector integration sites (IS) were retrieved on T-ALL blasts using a sonication-based PCR protocol named SLiM-PCR[4,14]. These analyses identified that blast cells comprised a single highly dominant clone with a vector integration site ~40 kb upstream of and in an antisense orientation to the transcription start site of *LMO2* (Fig. 2a)[7,8,10]. Insertion analyses revealed an overall polyclonal profile on whole and lineage hematopoietic cells purified from PB and BM up to 3 years post-GT (*n* = 10,332) (Fig. 2b–d, Supplementary Fig. 2a–c; Supplementary Table 1). The cell clone with the vector integration at the *LMO2* locus (*LMO2* clone, hereafter) was retrospectively found in mature PB CD4+ T cells since the first-year post-GT with a relative abundance of 6%, and its level clone remained stable until T-ALL occurrence. In addition, ISs retrieved by LiBIS-seq[14] from blood-derived cfDNA collected since the first month post-GT identified that the γ-RV IS of the malignant clone was detected starting from +31 months post-GT and its level progressively increased over time reaching 90% at the time of diagnosis (Fig. 2e, Supplementary Fig. 2d; Supplementary Table 1). This expansion was paralleled by a progressive increase in the amount of cfDNA post-GT which reached a maximum of 237,000 ng/ml at the time of T-ALL diagnosis, likely correlating with the leukemia outgrowth (Supplementary Fig. 2e)[14].

Using an integration-specific ddPCR, we tracked and quantified the abundance of the pre-neoplastic *LMO2* clone in PB-and BM-derived cells starting from one-year post-GT. This clone was particularly evident in the PB lymphoid compartment (3%) but also in CD34+ and CD56+ cells purified from the patient's BM at 12 months (0.03% and 0.03%, respectively), suggesting that it may have engrafted as a lymphoid or earlier progenitor persisting over time (Fig. 2f, Supplementary Fig. 2f, g). Notably, a distinct γ-RV IS within the *MECOM* proto-oncogene (Supplementary Fig. 2h)[7,11,12] was identified among the top 10 most abundant IS in whole-blood and BM cells, reaching highest level of abundance (26%) in BM-derived CD15+ cells at 36 months post-GT (Fig. 2c–e, Supplementary Fig. 2a–d, Supplementary Table 2, 3). Using an integration-specific assay, we detected a progressive increase of *MECOM* IS, reaching ~25% level of abundance in total DNA in CD15+ cells and whole BM and PB cell populations during chemotherapy cycles, while it completely disappeared starting from +3 months post-HSCT (Fig. 2g, Supplementary Fig. 2i).

To assess the functional influence of the γ-RV insertions leading to dominance, we quantified the expression of the *LMO2* and *MECOM* targeted genes. *LMO2* expression showed the highest value in P21 blast cells as compared to primary T cells from healthy donors and from a T-cell lymphoma that developed in a SCID-X1 patient (P9SCID-X1) consequently to a single IS located 10 kb upstream of the one described in this work (Fig. 2h)[13,14]. Interestingly, a progressive reduction in *LMO2* gene expression levels was observed upon serial passages of the blasts in xenograft models, suggesting that a progressive reduction in *LMO2* expression dependence for the leukemia outgrowth occurred upon serial transplantation (Supplementary Fig. 3)[23]. Moreover, we found that *MECOM* expression increased overtime in PBMC collected

**Table 1 | Clinical History, Treatment Characteristics and Outcome in ADA SCID P21 with T-ALL developing after γRV-GT**

| Variable | Patient Information |
|---|---|
| *Demographic characteristic* | |
| Gender, Race | M, Caucasian |
| Family history of malignancy | No |
| *Patient's features* | |
| Age at diagnosis (months) | 4 |
| *ADA* gene mutation | Missense c.455 T > C, Splice site c.478+6 T > C |
| Months of PEG-ADA before GT | 8 |
| *γRV-GT drug product characteristics and clinical course* | |
| Age at treatment (months) | 12 |
| Treatment frame | Named Patient Program (NPP) |
| Mean estimated total area under the curve busulfan concentration (ng/ml h) | 18132 (expected range 19200-22400) |
| Infused CD34+ cells (10^6/ kg) | 14.4 |
| VCN in transduced cells (copies/genome) | 1.8 |
| Quality tests[a] | Compliant |
| Nadir ANC (day) | $0.4 \times 10^9$/L (+12) |
| Neutrophil engraftment day § | +35 |
| Platelet nadir (day) | $94 \times 10^9$/L (+31) |
| *Post-GT immune reconstitution* | |
| Anti-CD3 and PHA proliferation at 1 and 3 years | Normal |
| IVIG discontinuation (months after GT) | 12 |
| Vaccination start (months after GT) | 15 |
| Vaccination response | Present for Chickenpox, Measles[c], Rubeola, Mumps, Tetanus, Diphteria, Hib, Pneumococcus; borderline for Pertussis; absent for HBV |
| *T-ALL diagnosis and treatment* | |
| Interval between GT and T-ALL diagnosis (age at diagnosis) | 4.7 years (5.75 years) |
| Clinical symptoms at onset | Hemorrhagic diathesis and weakness |
| Main laboratory findings | WBC $291.3 \times 10^9$/L (ANC $10.49 \times 10^9$/L, L $151 \times 10^9$/L, M $4 \times 10^9$/L), Plt $59 \times 10^9$/L, Hb 13.3 g/dl; AST 368 U/L, LDH 14000 U/L |
| Peripheral smear | Mainly immature lymphocytes (blasts 93%) |
| Flow-cytometry analysis (centralized AIEOP protocol) | CD45+++, CD1a−, CD2+++, sCD3−, cyCD3+++, CD4−, CD5+++, CD7+/−, NG2−, CD8−, CD22−, CD56−, CD99+++, CD10−, CD11a++, CD16−, CD19−, CD20−, CD24−, CD58−, CD13−, CD15−, CD33−, CD3 alpha/beta−, CD41a−, CD61−, CD65−, CD66b−, CD116−, CD117+, CD123−, CD133−, CD34 part-pos2, HLA-DR−, CyIg−, CD66c−, CD44++, TdT +/−, CD38 +/−, cyCD79a−, cyCD22−, CD11b−, CD3 gamma/delta−, MPO−, CD14−, CD64−, Lysozyme−, CLL-1−, Blasts 95% |
| Karyotype | 46, XY, del (6) (q21?) [8] / 46, XY [5]. Male karyotype with 62% clone of the cells examined with deletion of the long arm of a chromosome 6 with break points in 6q21 |
| γ-RV VCN/genome on PBMC | 1.18 |
| TCR Vbeta repertoire (FACS analyses) on non-malignant T cells | Most TCR Vbeta in normal range, except Vbeta2 (increased to 29.2%, normal range 4.03-23.48) and 4 Vbeta which were decreased |
| CNS status | CNS1 |
| Mediastinum status | No mediastinal mass |
| EGIL diagnosis | T-I/II, CNS1 |
| T-ALL treatment | AIEOP-BFM ALL 2017 protocol |
| Clinical course | No response to 2 days of steroids; cyclophosphamide, then started vincristine and daunorubicin (day +4); WBC decreased to $1 \times 10^9$/L (day +6) |
| Induction therapy outcome | Morphological complete remission obtained at the end of induction therapy |
| BM PCR-MRD day +33, TP1 | Marker 1: $5.6 \times 10^{-3}$; Marker 2: $3.9 \times 10^{-3}$ |
| BM PCR-MRD day +78, TP2 | Marker 1: $1.6 \times 10^{-3}$; Marker 2: $1.6 \times 10^{-3}$ |
| BM FCM-MRD after HR1 and HR2 | Neg |
| BM PCR-MRD after HR3 | $1 \times 10^{-4}$ |
| *HSCT characteristics and clinical course* | |
| HSCT | MUD not available. TCR αβ/CD19+ depleted HLA-haploidentical HSCT from the mother |
| HSCT conditioning regimen | TBI 990 cGy/TT 10 mg/kg, Fludarabine 160 mg/sqm |
| PTLD prophylaxis | Anti-CD20 mAb 200 mg/sqm |
| HSCT composition | WBC/kg $1.53 \times 10^9$ (viability 96%); CD34/kg $29.2 \times 10^6$; CD3/kg $21.4 \times 10^6$; TCR αβ/kg $0.069 \times 10^6$; TCR γδ/kg $21.2 \times 10^6$; NK/kg $66.8 \times 10^6$; CD20/kg $0.09 \times 10^6$ |

**Table 1 (continued) | Clinical History, Treatment Characteristics and Outcome in ADA SCID P21 with T-ALL developing after γRV-GT**

| Variable | Patient Information |
|---|---|
| Neutrophil engraftment day[b] | +12 |
| Platelet engraftment day | +11 |
| DLI (dose/kg, day post-HSCT, n of infusions) | $1 \times 10^5$, +37, 1 |
| aGvHD/cGvHD | None |
| T-ALL evaluation post-HSCT | |
| 1 mo F-U | BM FCM-MRD neg, PCR-MRD $< 5 \times 10^{-4}$ in 1 Marker |
| 2-14-29 mo F-U | BM FCM-MRD neg, PCR-MRD neg, LP neg (3 mo), 100% donor chimerism |
| Current F-U | 33 mo post-HSCT, alive, no signs of relapse, clinically well |

*ADA SCID* Adenosine Deaminase-deficient Severe Combined Immune Deficiency, *T-ALL* T-cell acute lymphoblastic leukemia, *γRV-GT* Gammaretroviral-Gene Therapy, *PEG-ADA* Polyethylene glycol-modified adenosine deaminase, *VCN* vector copy number, *ANC* absolute neutrophil count, *PHA* Phytohaemagglutinin, *IVIG* Intravenous ImmunoGlobulin, *Hib* Haemophilus influenzae type B, *HBV* Hepatitis B Virus, *WBC* white blood cells, *L* lymphocytes, *M* monocytes, *Hb* hemoglobin, *Plt* platelets, *AST* aspartate, aminotransferase, *LDH* lactate dehydrogenase, *sCD3* surface CD3, *cyCD3* cytoplasmic CD3, *TCR* T Cell Receptor, *FACS* Fluorescence Activated Cell Sorting, *CNS* Central Nervous System, *AIEOP* Associazione Italiana di Ematologia e Oncologia Pediatrica, *EGIL* European Group for the Immunological Classification of Leukemias, *PPR* Prednisone Poor Responder, *FCM-MRD* flow cytometry minimal residual disease, *BM PCR-MRD* Bone Marrow Polymerase Chain Reaction Minimal Residual Disease, *TP1* Timepoint 1, *TP2* Timepoint 2, *HR1, 2, 3* High Risk Blocks 1, 2, 3, *HSCT* Haematopoietic Stem Cell Transplantation, *MUD* HLA-matched unrelated donor, *TBI* Total Body Irradiation, *TT* thiotepa, *aGvHD* acute Graft-versus-Host Disease, *cGVHD* chronic GvHD, *PTLD* Post-Transplant Lymphoproliferative Disease, *mAb* monoclonal Antibody, *DLI* Donor Lymphocyte Infusion, *mo* month/months, *F-U* Follow-Up.
[a]Mycoplasma (PCR), cytokine-dependent growth, clonogenic test, microbiological control.
[b]First of three consecutive days with ANC ≥ 0.5 × 10⁹/L.
[c]Measles wild-type virus.

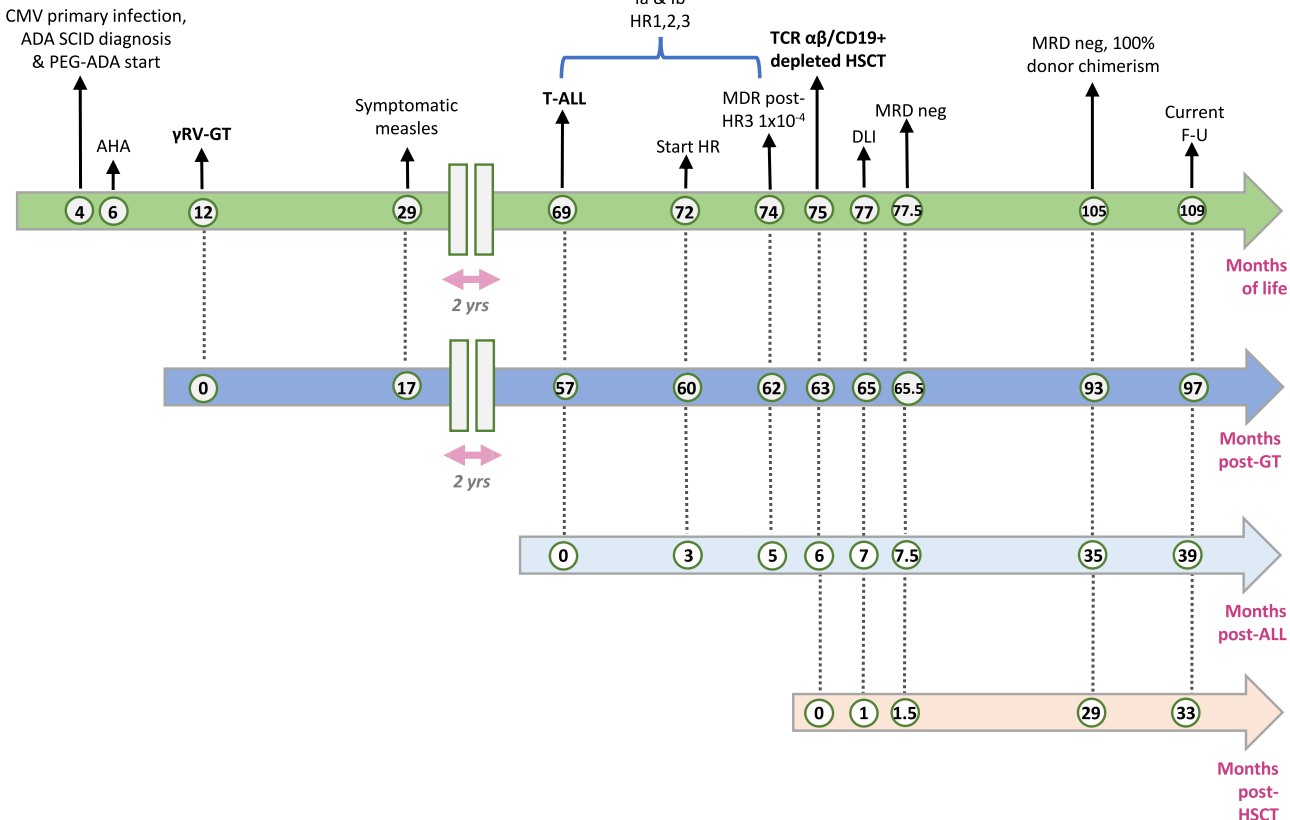

**Fig. 1 | Timeline of main clinical events of ADA SCID P21, having developed T-ALL after γRV-GT.** In the top green arrow, the main clinical events are reported as months of life, in the second blue arrow as months post-GT, in the third light blue arrow as months post-T-ALL onset and in the bottom salmon arrow as months post-allogeneic HSCT. Data are updated as of June 2023. The dashed lines between the arrows connect the same clinical events on different timelines. *Measles from wild-type virus. CMV: cytomegalovirus; ADA SCID: Adenosine Deaminase-Deficient Severe Combined Immune Deficiency; PEG-ADA: Polyethylene glycol-modified adenosine deaminase. AHA: Autoimmune Hemolytic Anemia. γRV-GT: Gamma-retroviral-Gene Therapy T-ALL: T-cell acute lymphoblastic leukemia Ia & Ib: Induction therapy a & b; HR1, 2, 3: High-Risk Blocks 1, 2, 3 MRD: Minimal Residual Disease HSCT: Haemato-poietic Stem Cell Transplantation DLI: Donor Lymphocyte Infusion F-U: Follow-Up.

at +60 and +62 months post-GT, reflecting the expansion of the *MECOM* IS detected by ddPCR (Fig. 2i) and highlighting, that the γ-RV integration was responsible for the transcriptional activation of *MECOM*.

We next investigated if epigenetic mechanisms could be responsible for the reduced ADA activity observed in blast cells versus those detected in PBMCs at earlier time points. Bisulfite sequencing revealed significantly higher levels of methylation in the CpG island of the

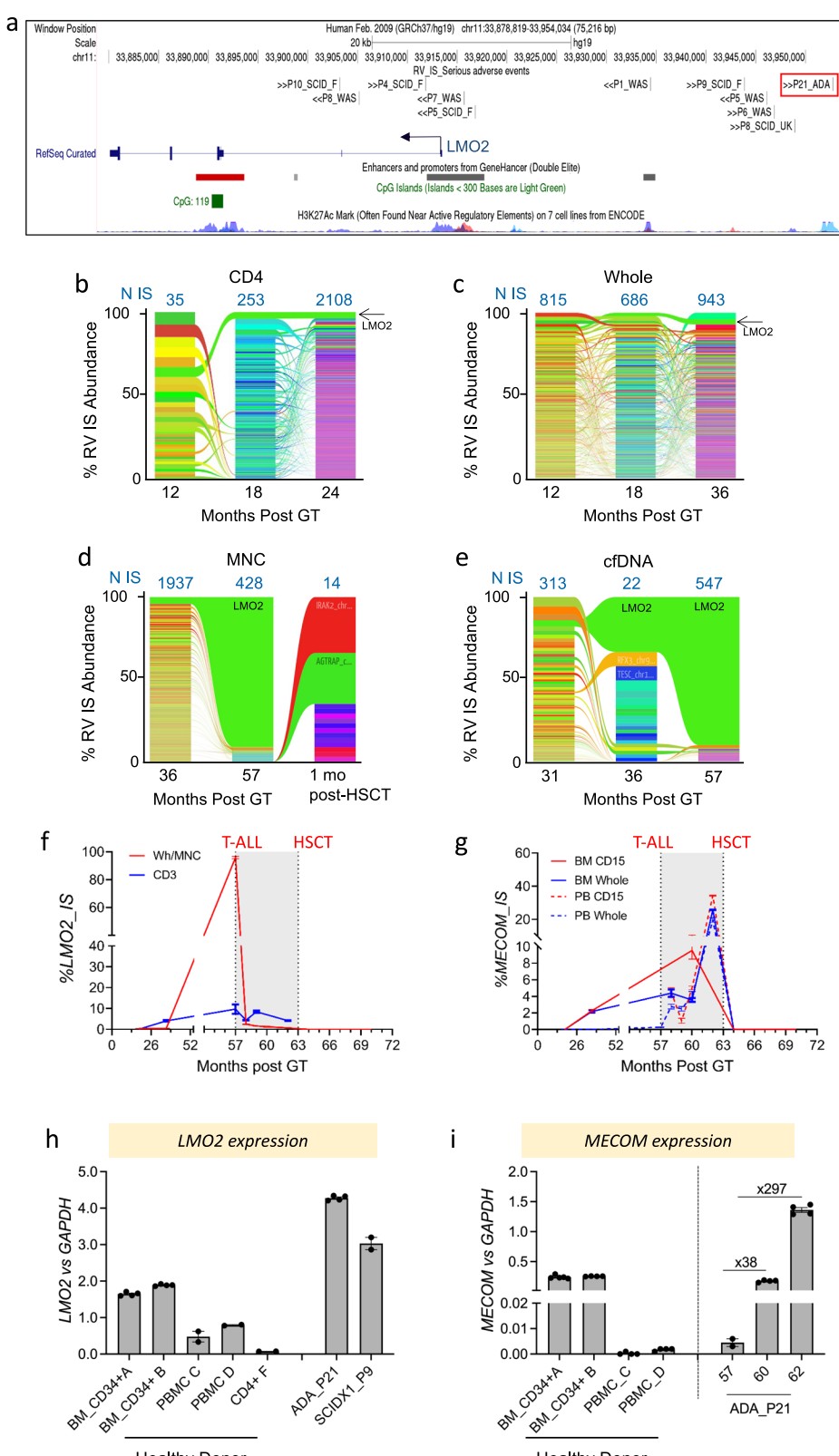

promoter region of the 5′ long terminal repeat (LTR) in blast cells (89.8% methylation level C per site) compared to CD4+ T lymphocytes collected at two years post-GT from the same patient, when transgene activity was higher (12.5%, Fig. 3a, b). In contrast, no methylation at CpG dinucleotides was observed in both samples within the enhancer region in the U3 region of the viral LTR. These data highlighted a

specific epigenetic fingerprint occurring at the vector promoter in the leukemic blasts.

Since the orientation and position of the vector integration suggest that the overexpression of LMO2 was caused by enhancer-mediated trans-activation mechanism, we performed in-situ Hi-C[24] on P21 blast cells to assess if the un-methylated enhancer sequences of the

**Fig. 2 | Distribution of vector integrations into *LMO2* and *MECOM* genes.**
**a** Genomic view of γRV IS close to *LMO2* retrieved from patients that developed serious adverse events as a consequence of vector-driven insertional mutagenesis. γRV IS retrieved from P21_ADA is highlighted by a red box. Chromosome, genomic coordinates and scale are indicated. Black lines refer to the position of the indicated γRV IS, black arrows refer to vector orientation. Patient_ID and disease are also indicated: UK, England and F, French SCID-X1 clinical trial. *LMO2* genomic structure is indicated by blue boxes and vertical bars that indicate exons; blue arrow indicates the start site and gene transcription. Gene regulatory regions such as CpG islands, Enhancer and Promoter, and histone methylation marks are indicated by the USCS genomic track. **b**–**e** Stacked bar plots showing the abundance of γRV IS (years, *x*-axis) in PB-CD4 (**b**), Whole (**c**), MNC (**d**) and cell-free DNA (**e**) samples collected at different time points (TP) post-GT. In each column, each γRV IS is represented by different colors, whose height is proportional with the number of genomes retrieved for that IS over the total (%IS Abundance, *y*-axis). Ribbons connect γRV IS tracked among consecutive TP. The number of unique IS retrieved from each TP is indicated in blue above the column. **f**, **g** Quantification of the relative abundance of the γRV close to *LMO2* (**f**) and *MECOM* (**g**) measured overtime by ddPCR. **h**, **i** Relative level of expression of *LMO2* (**h**) and *MECOM* (**i**) measured in the leukemic clone at the diagnosis. Gene expression levels were normalized to *GAPDH* expression. BM CD34+, PBMC and CD4+ cells from healthy donors are used as reference. *LMO2* expression levels measured in T-cell lymphoma developed in a SCID-X1 patient (P9) consequently to a single γRV insertion located 10 kb upstream of the IS described in this work (Fig. 2a). *LMO2* expression showed the highest value in P21 blast cells as compared to primary cells and T-cell lymphoma from P9SCID-X1. From f to i, data are presented as mean values ± SEM of technical replicate values. Source data are provided as a Source Data file.

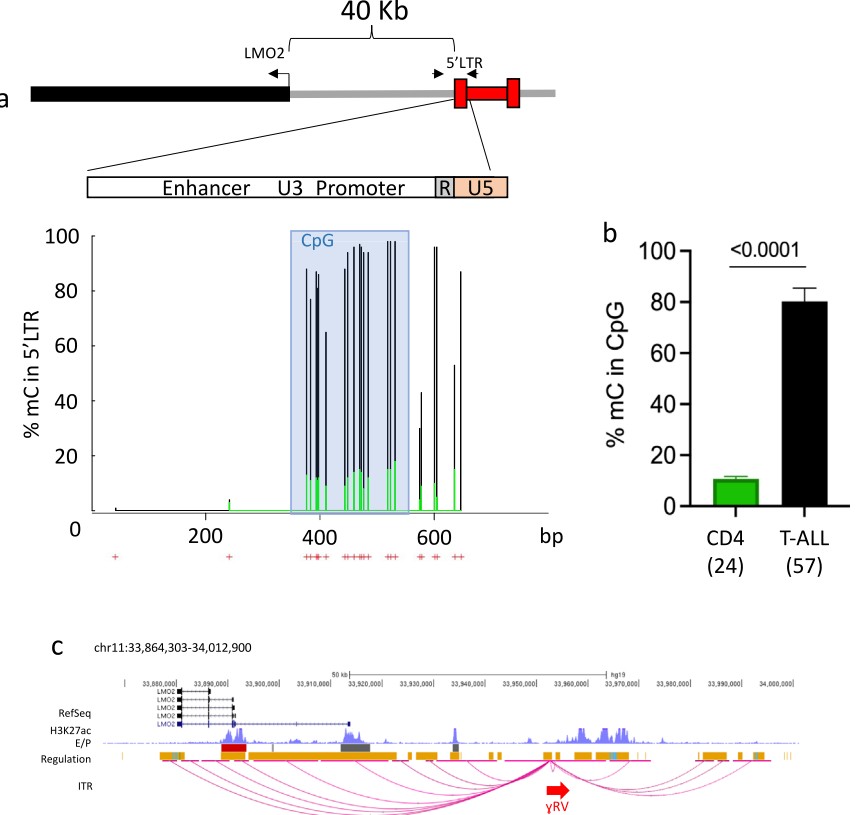

**Fig. 3 | DNA methylation of proviral RV LTR. a** Relative level of CpG methylation identified at the 5′ proviral LTR promoter region in leukemic blasts (black lines) and CD4 T cells collected from peripheral blood at 2 years post-transplantation. In leukemic blasts, the 5′ LTR region of the RV IS close to the LMO2 proto-oncogene was specifically analyzed. CpG islands of the promoter region within the proviral LTR is indicated in azur; (**b**) Overall level of methylated cytosine of the proviral RV LTR. A statistically significant difference between the methylation level at the proviral LTR sequence was observed in Leukemic blast compared to CD4 T cells collected 2 years post-GT. Data are presented as mean values ± SEM of the methylation level observed in the 23 CpG sites of the viral LTR. P value refers to two-tailed unpaired *t* test. c) γRV and host genome chromatin interactions at the LMO2 locus. A 149Kb genomic window is shown at the LMO2 locus. H3K27ac from K562 cells is shown as reference of active transcription and is displayed on top of the enhancer (gray) and promoter (red) bars track from Genehancer. Regulatory regions from Oreganno database (orange) and CTCF binding sites (blue) are shown. In-situ HiC sequencing reads harboring γRV and host genome sequences mapping at a distance <1000 bp were clustered into interaction peaks. Only peaks with at least 18 reads are shown. Horizontal pink lines indicate the span of the interaction peaks on the human genome. Interactions between the γRV integration site (red arrow) and human genome are shown as connecting arcs. The interaction data are merged from the patient assayed in two replicates and mouse xenograft BM and spleen tissues. Most of the γRV interactions target regulatory regions at the LMO2 locus, physically linking the integrated vector to the host genomic regulatory elements. Source data are provided as Source Data file.

5′ LTR were responsible for the upregulation of the *LMO2* proto-oncogene. This analysis identified the presence of strong interaction peaks stemming from the vector and pointing towards the enhancer and promoter regulatory regions of the *LMO2* locus, thus demonstrating a direct physical link between the integrated vector sequence with the host genomic regulatory elements of *LMO2* (Fig. 3c, Supplementary Fig. 4).

## Genetic abnormalities contributing to T-ALL oncogenesis
Whole genome, exome and RNA sequencing were performed on blast and BM-derived mononuclear cells collected before GT treatment to investigate whether genetic alterations and potentially dysregulated gene expression pattern and fusion transcripts may have contributed to the tumor formation.

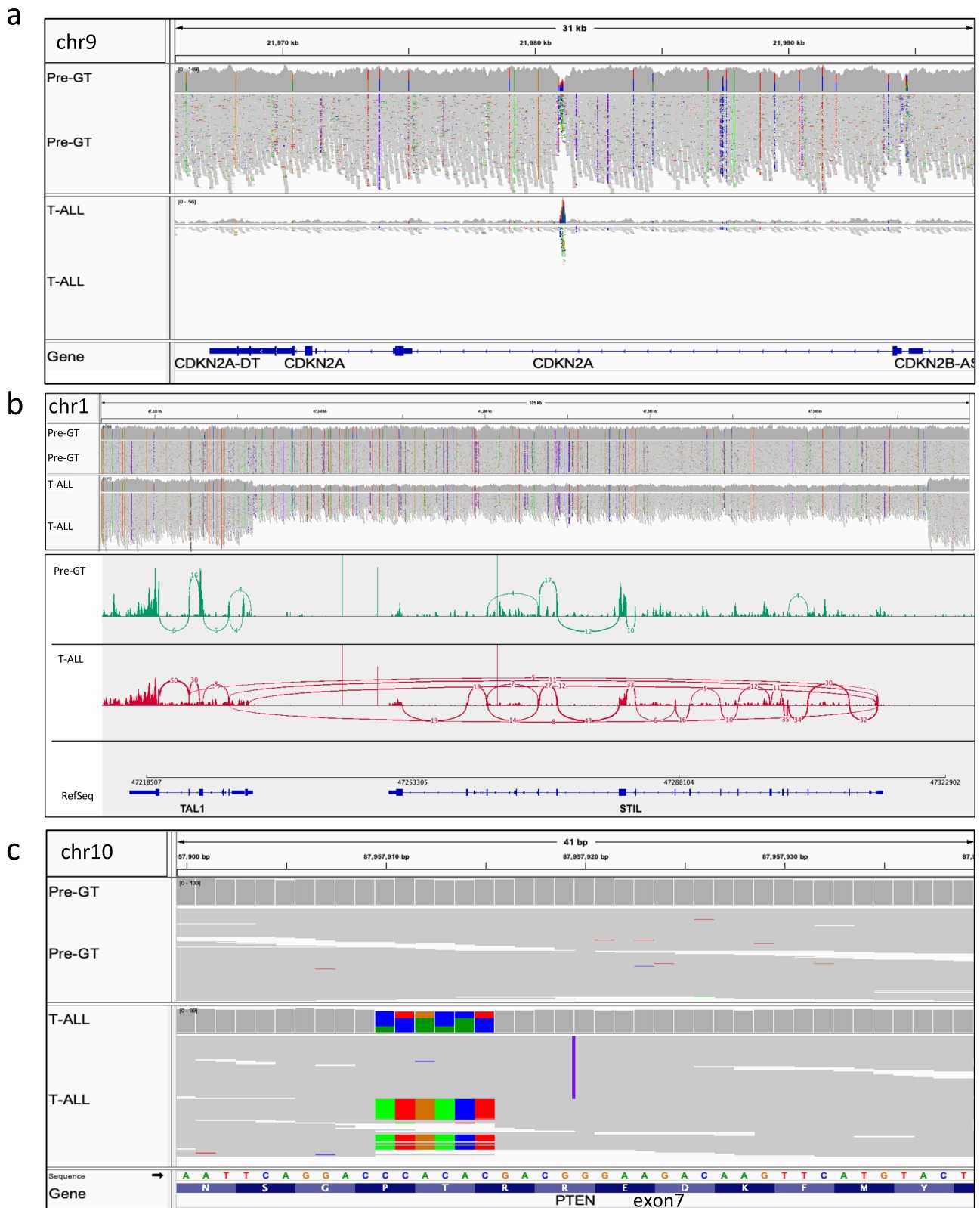

WGS confirmed the presence of a vector IS upstream of *LMO2* in blast cells. Several somatic variants (SVs) were also detected in the T-ALL when compared to pretreatment samples (Supplementary Table 4). Among the most relevant structural rearrangements, 3 large structural variants were detected: a single copy deletion of the long arm of chromosome 6 involving 327 genes, and two loss of heterozygosity (LOH) regions on chromosome 9 leading to a biallelic deletion of the tumor suppressor genes *CDKN2A* and *CDKN2B* and *MTAP* (Fig. 4a). Furthermore, a deletion on chromosome 1 predicted to lead to a *STIL-TAL1* fusion transcript was identified (Fig. 4b) and amplification of the *KANSL1-LRRC37A-ARL17* locus. Of note, all these genomic alterations have been previously described in T-ALL cases[25–27] and in SAEs occurring in γ−RV GT trials for other IEI[8,10,28]. Among the other alterations that had not specifically been reported

**Fig. 4 | Private genomic mutations in T-ALL and pre-treatment samples.**
**a** Genomic view of the loss of heterozygosity (LOH) mutation in the *CDKN2A* tumor suppressor gene(chr9) showing aligned reads and the corresponding genomic coverage for the entire region in the leukemic sample (bottom track) and in the pre-treatment sample (top track). **b** *STIL* gene deletion in the T-ALL sample. Coverage plot of the Pre GT (first panel) and T-ALL sample (second panel) is shown for the genomic region where *TAL1* and *STIL* gene are located. Coding regions of the entire *TAL1* and *STIL* gene are detailed in blue color at the bottom of the figure. Total RNA sequencing data showing for the pre-gene therapy (third panel) and T-ALL sample (fourth panel) the transcription of the indicated region. In each panel, the top section describes the read coverage along the genome with exons highly

expressed; the middle section shows the splice junctions with plus strands (red-colored) and minus strands (blue-colored). The thickness of the arches (blue or red bows) represents the number of spliced reads for a particular region. The lower part of each graph individual represents the alignment of reads. The *STIL-TAL1* fusion transcript results from the splice junction with plus strands (red-colored).
**c** Genomic view of the 2 distinct coding mutations in *PTEN* identified by the aligned reads in the leukemic clone (top track) showing consecutive mismatches (colored portions of the reads) or insertion (violet mark). These mutations were not visible in the pre-treatment sample (bottom track). Aligned reads with no base changes are colored in gray.

in T-ALLs, we documented chromosomal translocations at chromosomes 3, 5, 6, 17, 19 and X, and deletions at chromosomes 7 and 14 related to the rearrangement of T-cell receptor genes. Finally, exome sequencing revealed the presence of two *PTEN* variants in compound heterozygosity within the exon 7 a 6-nucleotide substitution and a dinucleotide insertion (Fig. 4c). These variants were predicted to cause premature termination of *PTEN*, likely altering the PI3K/AKT signaling pathway in the leukemic cells (Supplementary Fig. 5).

RNA sequencing in blast cells confirmed the overexpression of *LMO2*, the presence of the *STIL-TAL1* and *KANSL1-LRRC37A* fusion transcripts and numerous highly expressed genes associated with T-ALL[25,26] (Supplementary Table 4). Furthermore, dominant alpha/beta T cell receptor chains were also detected to be highly expressed, thus confirming the presence of a TCR clonotype in the leukemic cells. Of note, only about 13% of the total *ADA* transcript were of vector origin, whereas the remainder were from the germline mutant alleles. Next, we explored if any germline mutation was present in critical genes that may predispose the patient to the development of cancers[29]. By comparing genetic variants retrieved in blast with those retrieved from BM mononuclear cells archived before GT and somatic cells collected from buccal swab, we identified 21 non-silent germline mutations predicted to impact protein function (Supplementary Table 5). All these variants are present in heterozygosity, and many of them occurred in tumor suppressor genes and transcription factors that have a role in cancer development and T cell function, like MSH6, ARID1A, CARD11, CBL and SRC[30–33].

Single-cell RNA sequencing showed that a pre-GT CD34+ BM sample differed from the post-GT samples collected during the first 3 years after GT by a high proportion of lymphoid progenitors, some of which overlapped with the leukemia cluster (Supplementary Fig. 6a–d and Suppl. Data 1). This result may be related to the young age of the patient and/or disease-specific biology, before correction of the genetic defect, and is in line with our previous findings of increased frequency of lymphoid progenitors in ADA-SCID patients[34]. *LMO2* was highly and homogenously expressed across all leukemia populations, but also in monocytes and dendritic cells and the majority of CD34+ cells, even before GT (Supplementary Fig. 6e), making it an unreliable transcriptional marker to distinguish the leukemic clone from non-malignant progenitors. A module score composed of a cluster of genes identified from bulk RNAseq data accurately distinguished leukemia blasts from BM cells collected 2-3 years before leukemia diagnosis, arguing that the *LMO2* clone detected at low frequency in CD3+ T cells (~4%) and whole BM (~0.5%) at the 3 years timepoint was not directly related to the progression toward transformation (Supplementary Fig. 6f–h).

### Comparison of IS analysis with other ADA-SCID patients

ADA_P21 is the first patient that has developed a T-cell ALL among more than 75 patients (including our cohort) that have been treated worldwide with γRV HSPC-GT. Hence, we asked whether any specific pattern could be observed in the vector integration profile of this patient (P21 dataset, N = 10332) compared with the IS dataset obtained from ADA patients belonging to our cohort of patients treated in the

clinical development phase (Sr-TIGET ADA-SCID dataset, N = 52317 IS, N = 22 patients) (Supplementary Table 1)[18] and a recently published IS collection obtained from 10 ADA-SCID patients that received HSPC-GT following reduced intensity conditioning using a γ-RV with Myelo-proliferative Sarcoma Virus-derived LTR sequence, (MND)-ADA (US ADA-SCID study, N = 5150)[35].

As reported previously for γ−RV, IS retrieved from the different datasets were unevenly distributed across human chromosomes showing a strong preference for gene-dense chromosomal regions, TSS and CpG islands[21] (Fig. 5a, Supplementary Fig. 7a, b). An overall polyclonal pattern was observed and maintained overtime in SR-TIGET cohort, with few detectable expanded clones in some patients (relative abundance level >20%) (Fig. 5b, Supplementary Fig. 7c–d). *MECOM* and *LMO2* were the most frequently targeted genes among the three IS collection, however, *MECOM* was targeted at a significantly higher frequency in the cohort of patients treated with the γ-RV carrying the MND promoter ($p < 0.0001$ by Fisher's exact test) (Fig. 5c). In line with this result, common insertion site (CIS) analysis by the Grubbs test for outliers[36] showed that *MECOM* was a CIS only in the IS dataset from the US cohort of ADA patients, while *LMO2* appeared as a CIS in all the 3 different datasets (Fig. 5d), and resulted most frequently targeted in P21 compared to the other Sr-TIGET cohort of patients (Fig. 5c). Beside the targeting and selection of specific gene, we also investigated whether the gene classes targeted by the vectors were similar in the IS dataset. Gene ontology enrichment (GO) analysis identified a similar preference towards gene classes involved in T cell activation and differentiation, migration and proliferation, as observed in previous studies[20,21,35] (Supplementary Fig. 8a–c). However, the highest level of semantic correlation (0.89) (Supplementary Fig. 8d) was observed between the GO classes enriched in ADA_P21 and the other ADA-SCID patients belonging to the same clinical cohort (ADA-P21 vs US cohort of patients 0.83) (Supplementary Fig. 8d).

## Discussion

A report of long-term safety and efficacy of 43 patients with ADA-SCID who received retroviral ex vivo bone marrow-derived hematopoietic stem cell GT in the context of clinical development or named patient program and post-marketing has been recently published, documenting a safety profile of the real world experience in line with premarketing cohort[18]. Here, we report and extensively describe the single case of T-ALL occurring 57 months post-infusion in an ADA-SCID patient treated by γ-RV HSC-GT during clinical development of Strimvelis. As in most cases of vector-induced T-ALL, leukemia development was initiated by a single γ-RV insertion that occurred upstream to the *LMO2* proto-oncogene leading to its dysregulation. Our three-dimensional chromatin conformation study demonstrates a direct physical interaction between the integrated γ-RV enhancer sequences in the 5′ LTR and the transcription factor-enriched regulatory sites of *LMO2*, confirming that enhancer-mediated mechanisms were responsible for *LMO2* overexpression.

*LMO2* dysregulation has also been frequently observed in pediatric T-ALL consequently to chromosomal abnormalities[37,38]. *LMO2* has a critical role in hematopoietic cell development[39], it is expressed early

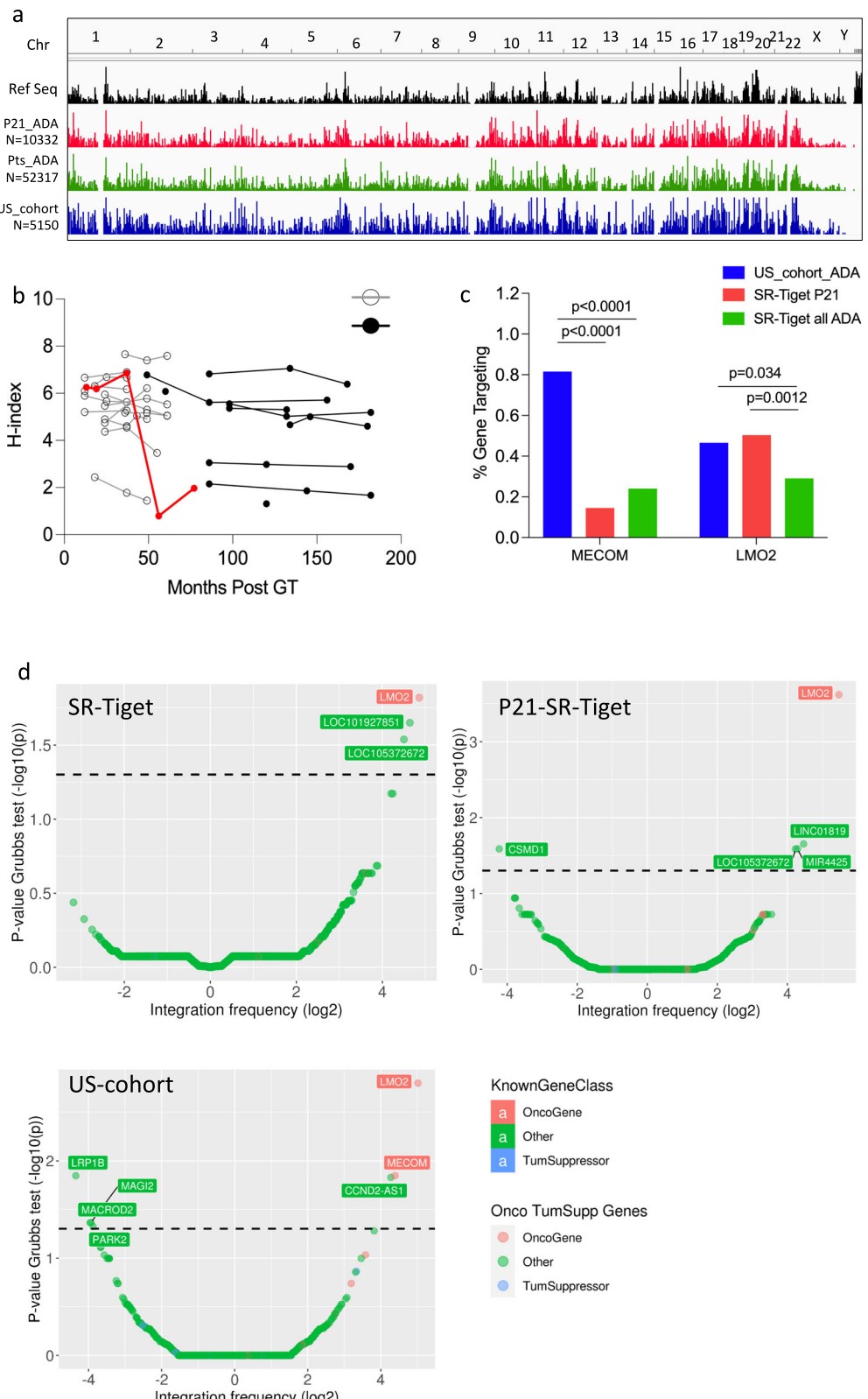

in hematopoiesis and its downregulation is crucial for T-cell maturation since its continuous expression is associated with a block in T-cell differentiation and enhanced thymocyte self-renewal[40,41]. Through integration studies and specific PCR, we found that the T-ALL-associated *LMO2* integration was detectable in mature CD4+ T cells and myeloid and BM CD34+ cells, indicating that the original transduced progenitor may normally differentiate for years after GT. It is

possible that the LMO2 clone retained the ability to downregulate *LMO2* overexpression to promote T-cell differentiation or that its overexpression levels were not sufficient to block cell differentiation. The presence of other 52 different IS within and close to the *LMO2* gene (Supplementary Fig. 9) further reinforce our hypothesis that the integration per se in this locus cannot prevent T-cell differentiation and was not sufficient to induce malignant transformation. One can

**Fig. 5 | Comparison of γRV IS retrieved from P21_ADA, all the other ADA patients from SR-Tiget and a US-cohort. a** Frequency distribution across the human genome of RefSeq genes (black bars) and of γRV retrieved in SR-Tiget P21_ADA (red bars), all other SR-Tiget ADA patients (green bars) and ADA patients from the US-cohort, blue bars. **b** Graph shows the H-index value (y-axis) measured over time from PB-derived cells of patients from the pivotal study and Long-Term Follow-Up (LTFU) protocol (filled circles) and from the LTFU Strimvelis (empty circles) IS datasets. In red is indicated the H-Index of the P21 patient. **c** Targeting frequency of *LMO2* and *MECOM* genes in the three different datasets; *P* value refers to Fisher exact test comparison on the targeting frequency in P21 and other

datasets. Source data are provided as a Source Data file. **d** Common insertion site (CIS) results for all ADA patients, patient ADA 21, and ADA patients from the US clinical trial. Results are represented as volcano plots in which each gene is represented with a colored dot with its integration frequency (x-axis, in log 2 scale) and the associated *p*-value from the Grubbs test for outliers (y-axis, in −log10). The dashed line is the threshold of significance for each group identified from the Grubbs test for outliers and Bonferroni correction (*p* = alpha value 0.05). Genes are colored according to their annotation from Uniprot database as Oncogene (in red), Tumor-Suppressor (in blue), or other (in green).

speculate that *LMO2* upregulation might have occurred in a pre-leukemic progenitor subclone only at a later time point favored by the promoter methylation at the 5′ LTR. CpG methylation of the viral LTR promoter has been described in transduced myeloid cells of X-CGD trial[11,12], while it has not been reported before in the context of *LMO2*-associated leukemogenesis. Although the expression of ADA does not directly confer a proliferative advantage to transduced cells, the residual low ADA activity in blast cells or the systemic detoxification deriving from the other transduced cells could be sufficient to guarantee cell survival and proliferation. On the other hand, the lack of methylation at the enhancer region of the LTR likely reflects a selective advantage dependent on vector-mediated trans-activation of *LMO2*.

Besides the integration at *LMO2*, T-ALL blasts were characterized by an extensive set of acquired rearrangements and intragenic lesions likely contributing to the multi-step oncogenesis process. Many of these events have been described in T-ALL and in serious adverse events occurring in γ-RV GT trials for other IEI[8,10,25,26,28]. Moreover, two unknown loss-of-function *PTEN* mutations were also identified in blast cells and likely contribute to clonal selection. RNA-seq and single-cell analyses also confirmed the expression of a gene signature associated with developing/proliferating T cells in the leukemic blasts. The absence of such specific T-ALL expression profile in mononuclear BM cells sampled 2-3 years before leukemia diagnosis suggested that at these time points and anatomic site the malignant transformation of the LMO2 clone was not yet achieved. An increasing proliferative ability acquired by the preleukemic clone can also be suggested by the progressive increase of the overall level of cell-free DNA and in the relative abundance of the *LMO2* IS starting from 31 months post-GT[14,42]. Furthermore, twenty-one non-silent germline mutations have been identified in P21. Although none of them have a known causal role in tumor formation, they might influence the timing and function of the targeted genes, thus predisposing for the T-ALL[29] in combination with the LMO2-activating insertion. T-ALL-specific mutations analyzed by deep sequencing on cfDNA samples as well as ddPCR specific for *PTEN* mutations performed on DNA samples collected over time up to 3 years post-GT, failed to detect the presence of any T-ALL mutations. It is possible that the approach was not sensitive enough or that the mutations occurred at later time points (36-57 months post-GT) for which samples were not available.

Our data also revealed the presence of a cell clone harboring an activating γ-RV insertion within *MECOM* gene, whose abundance increased slowly and steadily over time during chemotherapy. A progressive dominance of several *MECOM* clones was observed in X-CGD and WAS γ-RV clinical trials that finally led to acute myeloid leukemia[7,11,43]. Transcriptional activation of *MECOM/EVI1* is often associated with MDS and AML in humans, and results from the GT clinical trials underline a major role of MECOM deregulation in triggering clonal expansion. It is likely that in P21, the γ−RV activating insertion in *MECOM* conferred a fitness advantage to the cell in the course of the chemotherapy regimen[44]. We can speculate that the full hematopoietic replacement provided by the allogeneic transplantation prevented a potential progression of this pre-malignant clone towards a secondary myeloid neoplasm[44].

Immuno-deficient patients are known to be prone to hematological malignancies and 10 cases of lymphoma have been described in ADA-SCID patients undergoing ERT[45,46]. The incidence of malignancies post-transplant for SCID patients has been reported up to 2.3% in patients undergoing allogeneic HSCT[47,48]. Overall, 21 oncogenic events have been reported due to γRV-induced genotoxicity among the different HSPC-based GT trials for IEI2. In all T-ALL cases, the *LMO2* proto-oncogene was first activated by vector insertional mutagenesis and followed by many other secondary genome rearrangements that drove progression toward neoplastic transformation. However, the cumulative incidence of genotoxic events for ADA-SCID patients is the lowest that has been observed: 0.20 events per 100 years of observation, while in SCID-X1, X-CGD and WAS γ-RV GT clinical trials these values are much higher (4.1, 6.8 and 26.6, respectively)[2]. The lower frequency of leukemia in the ADA-SCID cohort might be influenced by several factors among which we can list the function of the transgene, the disease background and other yet unknown factors. Indeed, while IL-2Rγ and WAS function impacts lymphocyte proliferation and homeostasis[49], ADA is a detoxifying constitutive enzyme whose function has not been directly linked to cell proliferation and self-renew. Moreover, it has been shown that SCID-X1 background may constitute a risk factor for tumorigenesis per se because of the presence in the BM of SCID-X1 mutant mice of an expanded population of primitive progenitors highly prone to mutagenesis[50,51]. Similarly, it was shown that WAS deficiency increases tumor susceptibility and accelerates tumor growth[52]. Hence, compared to the other IEI, ADA background is associated with a lower risk of tumorigenesis. Other potential factors that have been proposed to favor the occurrence of delayed neoplastic transformation include the conditioning regimen, the vector dose in the drug product and the choice of enhancer/promoter sequences (Suppl. Table 6). In the SCID-X1 GT trial, the lack of conditioning limits the replenishment of progenitors from the BM, thus favoring the proliferation and selection of a reduced number of transduced cell clones[53]. Conversely, WAS patients developed T-ALL despite a higher dose of busulfan conditioning[7]. It is unlikely that the number of vector copies had a role in favoring transformation, since VCN was similar in SCID-X1 and ADA-SCID and no difference was found in VCN between P21 and the other ADA-SCID patients from our cohort. By comparing the integration profile of ADA patients treated in our center or in a US-cohort[35], we identified that while the *LMO2* locus was a CIS in all the datasets, *MECOM* is a CIS only when the MND-based γ−RV vector was adopted. These data confirmed that the enhancer/promoter sequences in the vector dictates the selection of genes close to the integration that may drive clonal expansion and, eventually, transformation. *LMO2* is one of the frequently targeted site in all patients' population presented here and in other γ-RV clinical trials[7,21,54-56], likely because this gene is a well-known hotspot for γ-RV integration due to the open chromatin open state of that region in CD34+ cells[21,57]. However, despite the high number of engrafted cells carrying such integration and undergoing some in vivo expansion, transformation does not necessarily ensue further proving the requirement for multiple cooperating events. To unravel what renders this patient unique as compared to the other ADA-SCID patients, we looked for the presence of germline mutations that could influence the risk for neoplastic

transformation in P21. It has been reported that approximately 8–10% of pediatric cancer patients harbor germline predisposing mutations[29]. We specifically identified 21 non-silent germline mutations that are predicted impact protein function. Some of these mutations occurred in known tumor suppressor genes and transcription factors that have a role in cancer development. Among those, missense mutations were found in five genes (MSH6, ARID1A, CARD11, CBL and SRC) which are involved in T-cell development and that previously were found altered in lymphoid and myeloid leukemia[30–33]. Although none of the variants found have been specifically described to have a causal role in tumor formation, it could be possible that they might have influenced the timing and function of the targeted genes, predisposing for the T-ALL development through various mechanisms (tumor intrinsic and/or immune-mediated) and in combination with the LMO2-activating insertion and the multiple acquired somatic alterations occurring in the tumor cells.

The main limitation of our study is that despite the extensive effort we have not clearly identified the factor(s) that can explain what renders this patient unique as compared to the other ADA-SCID patients. Indeed, several genetic, environmental, or constitutional characteristics of the individual may have contributed to the transformation event, and understanding the factors that led to this event in P21 is a challenging task[58].

After more than 20 years of follow-up of γRV-based HSC GT in the ADA-SCID cohort, this study shows that a sporadic T-ALL developed in one of our ADA-SCID patients. This adverse event was triggered by retroviral integration at the *LMO2* locus and further driven by the acquisition of a complex set of somatic mutations, some of which are well known to occur in T-ALL, thus recapitulating our current knowledge on disease development as experienced in other GT trials. The low frequency of such vector-related adverse events reported in γ−RV ADA-SCID GT as compared to the other immune deficiencies indicates that some disease-specific factors alleviate such risk. The risk/benefit balance remains favorable for Strimvelis in its approved indication in Europe[59] with the recommendation to continue long-term safety monitoring of treated patients.

## Methods
### Study participants
P21 was treated under a named patient program (NPP), before marketing authorization of Strimvelis, with approval from Institutional Ethical Committee of San Raffaele Hospital, Milan, Italy, and Italian competent authorities. The other ADA-SCID patients were enrolled in the pivotal study and LTFU protocol (registered at www.clinicaltrials.gov as #NCT00598481) and the LTFU Strimvelis registry (#NCT03478670) up to 3 years of FU. Complete clinical report and longer FU of all these patients, when available, have been reported elsewhere[18].

CD34+ cell purification from BM and transduction protocol, preconditioning with low dose Busulfan (0.5 mg/kg i.v. on 8 consecutive doses administered in 2 days (total dose 4 mg/kg), and AUC monitoring, have been reported elsewhere[18,22,60]. Blood and BM samples were obtained from all enrolled subjects after obtaining written informed consent from the parents or guardians following standard ethical procedures with approval of the Bambino Gesù Children's Hospital Ethical Committee and Institutional Ethical Committee of San Raffaele Hospital (TIGET06, TIGET09).

PBMC and plasma samples were obtained from patient P9 enrolled in the LTR-driven γRV-based SCID-X1 GT trial conducted between 1999-2002 at the French Hospital Necker–Enfants Malades, Paris. γc GT trial at Hôpital Necker–Enfants Malades, Paris. This P9 patient was previously reported as P8 in a previous ublications[61,62]. The protocol was registered under the local reference P971001, approved by the French Competent Authority (AFSSAPS) and the local Ethics Committee (Comité de protection des personnes of Hôpital Cochin, Paris, France).

P21 data monitoring and AE reporting was started from the date of GT and included 3 and 6 months, 1 year, 1.5 years, 2 years, 2.5 years, 3 years post-GT follow-up timepoints, with monitoring of full blood count and biochemistry, protein electrophoresis, immunoglobulins level, immunophenotype, VCN results; bone marrow morphology and karyotype were performed at 3 months, 1, 2 and 3 years follow-up; 4 years post-GT follow-up was performed at local hospital due to the pandemic. AE toxicity was classified using standard Common Terminology Criteria for Adverse Events (CTCAE) (version 4) criteria. According to EMA indications, all patients treated in the Clinical Development Program, Named Patient Program or with the commercial product will be monitored long term, with at least annual visits for the first 11 years and then at 13- and 15-year post-treatment, and follow-up will include a complete blood count with differential, biochemistry and thyroid stimulating hormone (#NCT03478670).

### Genomic analysis and VCN
VCN in cell subpopulations was used to assess engraftment. Genomic DNA was extracted from total PBMC using the Qiagen-midi DNA-Kit. From 2000 to 2012, the frequency of transduced cells and VCN were determined on genomic DNA by quantitative PCR analysis for NeoR vector sequences, normalized for DNA content[22]. Subsequently, the evaluation of VCN/genome was performed by ddPCR technology analyzing the LTR (long term repeated) vector sequence (Primer Fw: 5′-GGCGCCAGTCTTCCGATA-3′; Primer Rv: 5′-TGCAAACAGCAAGAGGC TTTATT-3′), normalized to a region of the human Telomerase gene.

### Retrieval and identification of vector integration sites
IS were retrieved using the Sonication Linker mediated (SLiM)-PCR and recently described[4,14]. Briefly, the SLiM-PCR procedure consists in the following steps: (i) fragmentation by sonication of the DNA (ii) ligation of the fragments to a linker cassette (LC) (iii) two consecutive rounds of PCR, to specifically amplify vector/cellular-genome junctions, by using primers annealing to the vector genome end (Long Terminal Repeats, LTR) and the LC. Primers contain DNA barcodes allowing univocal barcoding of all the SLiM-PCR replicates, and sequencing adapters that allow multiplexed sequencing on Illumina sequencers.

Sequencing reads were processed by a dedicated bioinformatics pipeline (VISPA2, repository: https://github.com/giuliospinozzi/vispa2)[63] that isolates the genomic sequences flanking the vector LTR and map them on the reference genome. Briefly, paired-end reads are filtered for quality standards, barcodes identified for sample de-multiplexing, vector sequences are trimmed from each read and the remaining cellular genomic sequence mapped on the reference Human genome (Human Genome_GRCh37/hg19 Feb. 2019) and the nearest RefSeq gene assigned to each unambiguously mapped integration site. VISPA2 eliminates sequences that: (a) do not have the entire LTR downstream the oligonucleotide used in the last amplification step; (b) are smaller than 19 nt, (c) do not map on the genome of interest, (d) map on multiple loci, (e) have a genome alignment spanning >1.2 kb, (f) whose paired ends map on different chromosomes or different genomic strands of the same chromosome. For the quantification of the abundance of each IS retrieved by genomic or cfDNA, we adopted the fragment estimate approach presented by Berry et al. and implemented in the R package as "*SonicLength*"[64] (available at https://cran.rstudio.com/web/packages/sonicLength/index.html). The abundance of each IS is determined by the number of different DNA genomes or fragments containing the same vector/cell genome junctions flanked by a genomic segment variable in size depending on the shear site position and that will be unique for each different cell genome present in the starting cell population. Therefore, the number of different shear sites assigned to an IS will be proportional to the initial number of contributing cells, allowing to

estimate the clonal abundance in the starting sample avoiding the biases introduced by PCR amplification.

Finally, we used a new R package, ISAnalytics to integrate the output files of VISPA2 and perform downstream analyses of IS[65]. This software removed the same IS in different independent samples, named collisions, using the same approach previously described[3] and samples containing a number of raw reads highly under-represented (3 fold less) than the average number of reads of the other samples in the pool (low-quality samples).

## Gene expression analysis

For the quantification of the RV IS identified in T-ALL blasts near LMO2 gene and within MECOM gene, custom-made locus specific ddPCR assays were designed (sequences available upon request). Ten to 150 ng of DNA were used for PCR amplification performed in triplicate and in a final volume of 20 ul. Abundance levels were measured by ddPCR using the QuantaLife ddPCR system using GAPDH levels (Hs00483111_Vic) as reference.

For gene expression analyses, total RNA was extracted from total hematopoietic cells and blasts using RNeasy purification kits (Qiagen) and reverse-transcribed with High-Capacity cDNA Reverse Transcription Kit (Applied Biosystems). cDNA was used as template for droplet digital quantitative -PCR. Expression levels were measured by ddPCR using the QuantaLife ddPCR system. ddPCR assays were used to assess gene expression of *LMO2* (dHsaCPE5026998) and *MECOM* (dHsaCPE5049452). The copies of tested genes were normalized to *GAPDH* (Hs00483111). Ten to 30 ng of cDNA were used for PCR amplification performed in duplicate and in a final volume of 20 ul.

In all ddPCR reactions, approximately up to 20,000 mono-dispersed droplets for each sample were prepared using the Quanta-Life droplet generator. The droplets were transferred to a 96-well PCR plate and amplified to endpoint in a standard thermal cycler (Bio-Rad) using the following conditions: 95 °C for 10 min, 40 cycles of 94 °C for 30", 60 °C for 60", and 98 °C for 10'. Plates were quantified in a QuantaLife droplet reader, and the concentrations of the targets in the samples were determined using QuantaSoft software.

## In vivo experiments

All in vivo experiments were performed upon approval by the San Raffaele Institutional Animal Care and Use Committee (protocol number 651), by the San Raffaele Ethic Committee (protocol AMLPDX, approved on November 3, 2017), and by the Italian Ministry of Health.

Blasts were engrafted into 4-week-old, non-irradiated male NOD-SCID γ-chain null (NSG) mice by tail-vein infusion. Engraftment was monitored weekly on 50 μL of peripheral blood by flow cytometry. Samples were stained in 100 μL of 1× PBS and 2% FBS plus the relevant mixture of antibodies for 10 minutes at room temperature (RT), using human CD45-PE-Cy7 (Clone HI30, Catalog. N° 304016, Lot N°B229089, 1:100), CD3-FITC (Clone SK7, Catalog. N° 344804, Lot N°B231398, 1:100) from BioLegend. The only anti-mouse antibody used is the pan CD45 PerCp5.5 (Clone 30-F11, Catalog. N° 103132, Lot N° B199699, 1:200) from BioLegend, (San Diego, CA, USA), utilized only for the in vivo experiments. After the incubation time, erythrocytes were eliminated by incubation in ammonium chloride potassium lysis buffer and samples were washed by centrifugation. For subsequent flow cytometry analysis, a first gate was set to discriminate between mouse and human CD45 cells and the absolute counts of leukemia blasts were quantified upon gating on the CD3 positive cells within the gate of human CD45 cells. The absolute count (cells/μL) was determined by the addition of count beads into each sample (Beckman Coulter). All the antibodies used in the flow cytometry experiments were from commercial vendors and they were validated for specificity to original targets by the manufacturers. The Certificate of Analysis is available from the manufacturers. They have been used according with manufacturer instructions provided in the data-sheets available at the

manufacturer's website at the reported link below or at the dilution specified above after in-house titering. Details are provided in the Reporting Summary. Mice were monitored three times a week. In agreement with the document approved by our Ethics Committee, the animals were euthanized when: the percentage of leukemic cells in peripheral blood was higher than 50%, a decrease in body weight higher than 20% was observed, and displayed signs of illness such as ruffled fur and hunched posture.

## Methylation studies

Bisulfite DNA conversion was performed using the EpiTect Bisulfite kit (QIAGEN) according to the manufacturer's instruction. For this procedure 500 ng of DNA was used. Converted DNA was PCR-amplified using locus-specific primes design using the MethPrimer software[66].

*For the amplification of the RV IS of the T-ALL we used the following primers*:

BS_First_Fw_LTR_TALL: 5'-AGCGGGGTTAACGATTATGGATTTAGT TG-3'

BS_Rw_LTR_TALL: 5'-GGAGGTAAGTTGGTTAGTAATTTATT-3',and for nested PCR:

BS_Inn_Fw_LTR_TALL: 5'-GGTTGATGTTATAATCGGATTGAGTATA TG-3'

BS_Inn_Rw_LTR_TALL: 5'-CTAAACAAAAATCTCCAAATCC-3'

*For the amplification of the RV LTR in CD4 we used the following primers:*

BS_First_Fw_LTR: 5'-AGATGGAATAGTTGAATATGGGTTAAA-3'

BS_Rw_LTR: 5'-GGAGGTAAGTTGGTTAGTAATTTATT-3' and for nested PCR:

BS_Inn_Fw_LTR: 5'-TTAGGGTTAAGAATAGATGGTT-3'

BS_Inn_Rw_LTR: 5'-CTAAACAAAAATCTCCAAATCC-3'

PCR was performed using 300 ng of converted DNA, the amplification was executed in a total volume of 50 μl using Taq polymerase (Qiagen). Nested PCR amplification was performed using 5 μl of the first PCR reaction. PCR were performed in a standard thermal cycler machine (Bio-Rad) and PCR conditions were the following: 95 °C for 5', 40 cycles of 95 °C for 45", 52 °C for 45", 72 °C for 45", and 72 °C for 2'. PCR fragments were agarose gel purified, cloned into the pCR4-TOPO plasmid (Invitrogen) and sequenced using the M13 universal primer to check for the specificity of the PCR reaction. Then, on selected amplified products Illumina barcodes were attached using TruSeq Nano DNA LT Sample Prep Kit (Illumina), libraries were then pooled and sequenced using an Illumina MySeq platform.

Next-Generation Sequencing reads were then aligned against a reference sequence of gRV vector. Sequenced reads were mapped to the viral genome using the Bismark algorithm (bismark v0.22.3, bow-tie2 v2.2.6[67]). Next, methylation calling (bismark --non_directional --genome <genome_folder > −1 <mates1 > −2 <mates2 > ) and extractor (bismark_methylaton_extractor -p −merge_non_CpG −bedGraph <file-names > ) were adopted to identify for each Cytosine the methylation results. Lastly, bedgraph-outputs have been produced and plotted with the R library "Sushi".

## Whole genome library preparation and analyses

Libraries for whole genome and cfDNA sequencing were prepared using the TruSeq DNA PCR-Free LP kit (Illumina) according to the manufacturer's instructions and starting from a 1000 ng of input genomic DNA material per sample. Sample libraries were pooled together and sequenced on the Illumina NovaSeq S4 using symmetric 150 bp PE sequencing. Paired reads were mapped on GRCh38 using the Isaac aligner[68] (Illumina, 2014). SNVs and small indels were called following GATK Best Practices[69]. Somatic SNV and small indel variants were produced by means of "Strelka"[69]. The "Manta" procedure[70] was used to identify structural variants (SV), defined as genomic rearrangements that effect more the 1 Kb. Copy number variants (CNV) were discovered by applying the "Canvas" procedure[71]. Final variant

annotation was done with the help of the Illumina Annotation Engine. Further analyses of raw data have been performed by using the software package R.

For the identification of T-ALL specific genetic alterations on cfDNA, paired reads were mapped with bwa-mem2 on a custom human GRCh38 genome where 11 T-ALL specific genomic alterations identified by the 100X Whole Genome Sequencing (plus 150 bp before and after the mutation events) were added as extra chromosome. Duplicate reads were then removed using Samtools (v1.16.1) and the coverage has been evaluated with Samtools depth for the custom sequences and with Samtools coverage for the standard GRCh38 using the option -q 1, to consider only reads that were correctly mapped. The 11 T-ALL-specific genetic rearrangements that we look for in cfDNA samples were: chr1: STIL rearrangement; chr14: TRAV21, chr7: TRGJ2_1, TRGJ2_2, TRBV4-1, TRBV20-1, and chr14 TRAV27, indicative of TCR rearrangements; chr6, deletions affecting more than 300 genes; chr9: leading to MTAP and CDKN2A deletions and chr14: GHV3-22 rearrangements (for more details refer to Suppl. Table 3).

## Exome library preparation and analyses

Genomic DNA was quantified using the Qubit 2.0 fluorimetric Assay (Thermo Fisher Scientific) and sample integrity, based on the DIN (DNA integrity number), was assessed using a Genomic DNA ScreenTape assay on TapeStation 4200 (Agilent Technologies). Libraries were prepared from 100 ng of total DNA using NEGEDIA OncoHaemo (NEGEDIA srl) which included library preparation, target enrichment using a Hematological specific probe set, quality assessment and sequencing on a NovaSeq 6000 sequencing system using a paired-end, 300 cycle strategy (2 × 150) (Illumina Inc.). Variant calling and annotation for the exome sequences were performed using previously published methods[72,73]. Briefly, Circulating leukocytes and saliva DNA were enriched using SureSelect All Exons v7 (Agilent) kit for exome sequencing. Raw sequence data were processed and analyzed following GATK Best Practices[69]. SnpEff v.5.0[74] and dbNSFP v.4.2[75] tools was used for known disease variants annotation (ClinVar), variant functional annotation, as well as for in-silico prediction of impact (CADD) v.1.6[76], Mendelian Clinically Applicable Pathogenicity (M-CAP) v.1.3[77] and Intervar v.2.0.1[78]. Population frequencies were annotated from both gnomAD database v2.1.1 and in-house database (~3000 exomes).

## RNA library preparation and analyses

Whole transcriptome sequencing was performed in samples from patient blast cells and from BM-derived mononuclear cells. Depending on availability, from 600 to 1000 ng of DNA-digested RNA were used for library preparation. Libraries were prepared using the TruSeq Stranded Total RNA LP kit (Illumina) according to the manufacturer's instructions. Sample libraries were then pooled and sequenced on the Illumina NovaSeq S4 using symmetric 150 bp PE sequencing. The Illumina® DRAGEN RNA pipeline performs Next Generation Sequencing (NGS) secondary analysis of RNA transcripts (Illumina, 2014). The RNA pipeline is based on multiple operating modes, including reference-only alignment and annotation-assisted alignment with gene fusion detection. Paired FASTQ files were used as input files. The gene fusion module leverages the DRAGEN RNA spliced aligner to perform split-read analysis on supplementary (chimeric) alignments to detect potential breakpoints. The Cufflinks Assembly & DE workflow performs the following functions to explore the differential expression of novel and reference transcripts. Results have been analyzed and filtered by using the software package R[79].

## Generation of Hi-C data and analysis

18×10^6 tumor cells from P21_ADA patient were purified with Dead Cell Removal Kit (Miltenyi Biotec 130-090-101) according to the manufacturer's instruction reaching a viability of 92%. Two aliquots of 4×10^6 purified tumor cells and FACS sorted tumor cells derived from spleen and bone marrow of xenotransplants were subjected to in-situ Hi-C with the Arima Hi-C kit (Arima, San Diego, CA, USA) following manufacturer instructions. Briefly, PBS-resuspended cells were crosslinked with 1% formaldehyde at room temperature for 10 minutes and reaction was stopped following user's instruction. For each generated HiC library, two tubes with 1.2 to 1.4 ug of library were sonicated by Covaris E220 ultrasonicator at the following conditions: 10% Duty factor, 200 cycle, 140 peak, 63 seconds. Biotin-enriched fragments were size-selected with an average size of 400 bp and subjected to end-repaired with the NEB Next Ultra-II kit (E7645L) and tagged with different Illumina DNA adapters. Libraries were amplified with 7 cycles following KAPA HyperPrep PCR conditions (07962347001) and verified by q-PCR with KAPA-Q-PCR reagents (07960140001). Libraries were sequences by Illumina NGS platform with paired-end sequencing at 300 cycles to reach a sequencing depth of 40x to 50x.

**HiC analysis.** Hi-C datasets were analyzed with the Juicer platform[80]. We created a custom reference genome starting from GRCh37/hg19 and introducing an extra chromosome with the GIADA vector sequence. Fragments were computed using the Digester function of HiCUPand results were adopted to the correct input format for Juicer with a custom script. Read pairs were aligned against this custom genome by using bwa (version 0.7.17)[81] exploiting the "mem" algorithm with default parameters for achieving the best accuracy. Considering paired alignments, duplicates are removed and read pairs that align to three or more locations are filtered out in a separated file. The catalogue of contacts obtained using this approach is then used to create two distinct contact matrices, the first considering all the alignments and the second discarding read-pairs with a MAPQ mapping quality <30. To create these matrices, the linear genome is partitioned into loci of a fixed size, or "resolution", (from 2.5 Mb to 5 kb) and these loci correspond to the rows and columns of the contact matrices. Each entry in a matrix reflects the number of contacts observed between the corresponding pair of loci during the Hi-C experiment. Downstream, Juicer provide some statistics to establish the quality of the experiment, such as the distribution of inter and intra chromosomal contacts and the percentage of short range (<20Kb) and long-range (>20Kb) interactions in datasets.

Due to factors such as chromatin accessibility, in Hi-C experiments certain loci are observed more frequently than others and therefore specific normalizations were applied by Juicer to correct these biases. The available options to normalize contact matrices include the widely used original normalization scheme proposed by Lieberman-Aiden et al[24], in which the entries in the contact matrix are divided by the average contact probability calculated genome-wide for loci at the same distance. The contact matrices generated in this way, using different resolutions and different normalization approaches, are stored efficiently in a compressed file format that is designed to facilitate all subsequent computations.

**TADs identification.** Starting from the HiC matrices generated with Juicer, we used the HiCexplorer tool to calculate topologically associated domains (TADs)[82]. To this aim, we applied the following steps: (i) converted each of the HiC matrices derived from Juicer with the "hicConvertFormat" from Hicexplorer tool generating the "cool matrix" format without normalization; (ii) normalized the matrix with the "hicNormalize" function (mode= smallest); (iii) applied the "hicCorrectMatrix" with KR correction, to balance the matrices using a fast balancing algorithm introduced by Knight and Ruiz[83]; iv) calculate TADs for each sample with "hicFindTads" with correction for multiple testing with FDR.

For P21 Rep1 TAD ID (ID_0.01_10922) is identified on chr11, 33865000-34010000 (145 Kb); for P21 Rep2 TAD ID (ID_0.01_11331) is identified on chr11, 33875000-34010000 (135 Kb); for mouse BM sample TAD ID (ID_0.01_11830) is identified on chr11, 33875000-

34010000 (135 Kb); for mouse Spleen sample TAD ID (ID_0.01_12086) is identified on chr11, 33880000-34010000 (130 Kb).

**Vector interaction analysis.** Mapping coordinates of reads with a paired end landing on the vector and the other pair mapping on the human chromosomes (chimeric) provides information about the viral insertion sites and the contacts established by the GIADA vector with surrounding chromatin. To analyze the distribution of these chimeric reads we specifically extracted them from the catalogue of contacts using the Linux command "awk". Using the mapping coordinates on the human chromosomes, we created bed and bedgraph files in order to visualize the results and integrating them with information about the known vector insertion sites. Moreover, we identified interaction peaks (ITRs) stemming from the vector by binning individual interaction that were at a distance >1Kbp for individual samples or 500 bp when the reads of the 4 datasets were merged. We assigned a score to each interaction peak, represented by the number of individual reads comprised within the peak-interval. To identify bonafide strong vector-derived ITRs over background we excluded the peaks with a score below the 10% of the total sequencing reads for individual samples, or a score below 1% when the four datasets were merged. Each interaction peak was associated with the vector insertion site and formatted as bigInteract file to display pairwise interactions as arcs connecting the vector Integration and the surrounding genomic regions.

**Single-cell library preparation and analyses**
Single cells were suspended in phosphate-buffered saline containing 0.04% bovine serum albumin, filtered using 40 um cell strainer (Biologix), and their concentration was evaluated at LUNA-II™ Automated Cell Counter (Logos Biosystems). The cell suspension was loaded onto the Chromium Single Cell G Chip Kit (10x Genomics) and run on the Chromium Single Cell Controller (10x Genomics) to generate single-cell gel beads emulsion, according to the manufacturer's protocol. The single-cell 3′ Library and Gel Bead Kit V3.1 (10x Genomics) were used to generate cDNA and the final libraries. The cDNA quality was assessed using High sensitivity D5000 screentape on Agilent 4200 TapeStation system (Agilent Technologies). The quality of libraries was assessed by using screen tape High sensitivity DNA D1000 (Agilent Technologies).

Finally, the libraries were sequenced on Novaseq6000 sequencer (Illumina) according to the manufacturers' specifications. Sequenced libraries were de-multiplexed and processed by Cell Ranger Single-Cell Software Suite (v6.0.1, 10X Genomics) using GRCh38 reference genome and gene annotations provided by the manufacturer (GRCh38-2020-A). We sequenced samples on BM cells at the time of T-ALL diagnosis, before gene therapy (CD34+ cells) and during the first 3 years after gene therapy, up to 21 months before leukemia onset (CD34+ and mononuclear cells). Quality control was performed to discard bad-quality cells and samples according to the number of expressed genes (>300 and <5000) and percentage (<15%) of mitochondrial genes expressed by each cell. For the leukemia sample, we first removed putative contaminants by discarding cells positive for myeloid or erythroid markers. Then, we performed doublets removal on all samples by using the cxds_bcds_hybrid function in scds R package[84]. Outlier cells with doublets score greater than $Q3 + 1.5 * IQR$ were classified as doublets and removed from the dataset. Data analysis workflow was performed with the R package Seurat (v3.3.2). In detail, counts were log-normalized and scaled for a factor of 10,000, followed by the selection of the top 20% most variable genes for downstream analysis. Cell cycle scores were assigned with the Cell-CycleScoring function using the reference gene lists included in the Seurat package[85]. Differences in cell cycle were defined as the difference between S phase and G2M phase module scores. Data were scaled and regressed out for UMI count (nCount_RNA), percent of mitochondrial genes (percent.mt) and cell cycle difference (cc.difference). Integration of multiple samples was performed by using the R package

Harmony (v1.0)[86] using orig.ident as covariate variable. UMAP dimensionality reduction and clustering were performed using the top 20 Harmony components. Clusters of cells were identified by a shared nearest neighbor (SNN) algorithm, using the classic implementation of the Louvain approach for modularity optimization. A clustering resolution of 1.8 was used for downstream analysis. Marker identification was performed with FindAllMarkers function setting a logFC threshold of 0.25. Genes were considered markers of a specific cluster, if their p-values were <1e10⁻⁶ (Wilcoxon Rank Sum test). Gene signature were evaluated by using the AddModuleScore function provided by the Seurat package

**Statistical analysis**
Statistical analyses were performed with GraphPad Prism 8.0 and R (version 3.5 or 4). Statistical significance for each CIS was established using the Grubbs test for outliers, as previously described[36]. Briefly, for each IS dataset, the targeting frequency of each gene was computed considering the number of IS landing in the gene body ± 100kbp and then normalized by the gene length. After the log2 transformation of the gene distribution frequency, the Grubbs test for outliers allows us to identify genes with a targeting frequency significantly higher than the average observed frequency.

GO has been realized using R packages for GO "clusterProfiler", the annotation DB "org.Hs.eg.db", "msigdbr". Semantic similarity has been done with the R package "GOSemSim"64. Feature annotations have been realized with the R packages "ChIPseeker" and database "TxDb.Hsapiens.UCSC.hg19.knownGene" (up-set plot) and the closest genes have been annotated with RefGene table (UCSC database hg19). Circos plot generated by the R package "circlize". The list of cancer genes has been obtained from the curated UniProt database (UniProtKB/Swiss-Prot, https://www.uniprot.org/). No data were excluded from the analyses. The Investigators were not blinded to allocation during experiments and outcome assessment.

## Data availability
All data supporting the findings of this study are available within the paper and supplementary files. Because of the small number of participants in the studies and potential for identification, individual patient data beyond what is included in the manuscript will not be available. Requests of additional information should be addressed to aiuti.alessandro@hsr.it and will be shared with Fondazione Telethon (the sponsor and Strimvelis license holder) R&D Director, to verify if the request is subject to any intellectual property or confidentiality obligations. Criteria for request evaluations will be: scientific merit of the request/ intellectual property restrictions/data transfer agreement. The timeline of response will be from 2 to 4 weeks. Source data are provided with this paper. Any data that can be shared will be released via a Material Transfer Agreement. Source data are provided with this paper.

## Code availability
All molecular analyses have been performed using dedicated software as indicated in the material and methods section. Sequencing reads were processed by dedicated bioinformatics pipeline (VISPA2, repository: https://github.com/giuliospinozzi/vispa2)[63], the fragment estimate approach presented by Berry et al. and implemented in the R package as "SonicLength"[64] for the quantification of the abundance of each IS (available at https://cran.rstudio.com/web/packages/sonicLength/index.html). Finally, we used a new R package, ISAnalytics to integrate the output files of VISPA2 and perform downstream analyses of IS[65].

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

## Acknowledgements

This work was supported by Telethon Foundation A.A. and E.M., the Italian Ministry of Health (GR-2018-12367860 to L.V. and D.Ce.), and the Else Kröner-Fresenius-Stiftung, Germany (EKF prize for Medical Research 2020 to A.A) and the Associazione Italiana per la Ricerca sul Cancro (IG #22197 to L.V.). This work was supported by grants awarded

by Fondazione AIRC per la Ricerca sul Cancro (Special Project 5 × 1000 n. 9962 to F. Locatelli, and IG 21614 and 28768 to M.T.), and Ministero dell'Università e della Ricerca (PRIN 2017WC8499_004 to F. Locatelli, and FOE 2019 and 2020 to M.T.). This work was supported by Italian Ministry of Health (Piano Operativo Salute Traiettoria 3, "Genomed"), to D. Ca. We wish to acknowledge the physician and nursing team of the Pediatric Immunohematology Unit, Stem Cell Transplant Program of the IRCCS San Raffaele Scientific Institute, for their professional care of patients during hospitalization and visits; Fondazione Telethon "Just Like Home" Program, Stefano Zancan and all SR-TIGET Clinical Trial Office, Matias Soncini and all SR-TIGET clinical lab for their support; the team of AGC Biologics (formerly MolMed) for manufacturing the vector and medicinal product. We thank Anna Manfredi for technical help. We acknowledge Claudio Bordignon, Giuliana Ferrari and Maria Grazia Roncarolo for helpful discussion on the clinical case. We would like to thank all the nurses and medical staff members of the Bambino Gesù Children's Hospital for their care of the patient during the chemotherapy treatment and HSCT procedure of the patient. Finally, we thank Don Kohn, Rick Bushman and John Everett for providing access to the published IS sequences obtained from ADA-SCID patients that received HSPC-GT transduced with γ-RV- (MND)-ADA vector[35].

## Author contributions

D.Ce. conceived and developed the project, performed experiments and wrote the manuscript; M.P.C. contributed to the study design, patients' follow-up, data collection, interpretation and manuscript writing. F.G, G.S., G.P performed bioinformatics analyses on γ-RV integration sites and part of the omics data under the supervision of A.Ca.; M.V. and I.M. performed and analyzed Hi-C data; M.Ba. and B.G. analyzed and interpreted the scRNAseq data; L.R., F.B., P.G. provided technical support for most of the experiments; P.M, R.C, M.M, F.Ba, F. Tu, Mi.Ca, L.S, L.V, L.P, C.Ca contributed to patient treatment, follow-up and data collection; C.F., contributed to data collection and analysis; S.Gi., F.D., S.S., E.D. and L.V. performed and analyzed experiments with patients' cells; E.S. and Ma.Ca. provided the clinical sample from SCID-X1 P9; M.Sm performed and supervised part of the analyses on WGS and RNA-seq samples; F. Ca performed biochemical analyses of ADA; S.P, A.Ci, A.B, and M.T generated, analyzed and provided interpretation of the WES and WGS data; D.Ca and M.C. oversaw the exome and single-cell library preparation, analysis and interpretation; S.R. performed exome library preparations and C. Co. performed exome data processing. F.Ci, B.G, M.T. and L.N. critically reviewed the manuscript; E.M. supervised the project, provide interpretation of Hi-C data and critically reviewed the manuscript; F.L. and A.A. contributed to the study design, data collection and interpretation, manuscript writing and provided overall supervision. D.Ce and M.P.C. contributed equally to this work. E.M, F.L. and A.A. authors jointly supervised this work.

## Competing interests

The San Raffaele Telethon Institute for GT (SR-TIGET) is a joint venture between the Telethon Foundation and Ospedale San Raffaele (OSR). GT for ADA-SCID was developed at SR-TIGET and licensed to GlaxoSmithKline (GSK) in 2010. The treatments under Named Patient Program and Hospital Exemption were provided free of charge by GSK. Strimvelis Marketing Authorization in Europe occurred in 2016 (under GSK holding) and then transferred to Orchard Therapeutics (Netherlands) B.V. in 2018. Following disinvestment by Orchard, Fondazione Telethon became the Strimvelis Marketing Authorization Holder in July 2023. The product is also currently licensed in Iceland, Norway, Liechtenstein, and UK. A.A: was the PI of pilot and pivotal SR-TIGET clinical trial of GT for ADA SCID. A.A. is a member of the Committee for Advanced Therapies (CAT) and his views are personal and may not be understood or quoted as being made on behalf of the European Medicine Agency (EMA). M.P. Cicalese is the PI of the Strimvelis Registry, RIS and RMMs studies. Davide Cacchiarelli is founder, shareholder, and consultant of NEGEDIA srl Sara Riccardo, Chiara Colantuono are employees of NEGEDIA srl. All other authors declare no other competing interests.

## Additional information

[1]San Raffaele Telethon Institute for Gene Therapy (SR-Tiget), IRCCS San Raffaele Scientific Institute, Milan, Italy. [2]Paediatric Immunohematology and Bone Marrow Transplantation Unit, IRCCS San Raffaele Scientific Institute, Milan, Italy. [3]Università Vita-Salute San Raffaele, Milan, Italy. [4]IRCCS Bambino Gesù Children's Hospital, Rome, Italy. [5]Molecular Genetics and Functional Genomics, IRCCS Bambino Gesù Children's Hospital, Rome, Italy. [6]Department of Oncology and Molecular Medicine, Istituto Superiore di Sanità, Rome, Italy. [7]National Research Council, Institute for Biomedical Technologies, Segrate, Italy. [8]Immune and Infectious Diseases Division, Research Unit of Primary Immunodeficiencies, Academic Department of Pediatrics, IRCCS Bambino Gesù Children's Hospital, Rome, Italy. [9]Immunogenetics, Leukemia Genomics and Immunobiology Unit, Division of Immunology, Transplantation and Infectious Diseases, Ospedale San Raffaele Scientific Institute, 20132 Milan, Italy. [10]Telethon Institute of Genetics and Medicine (TIGEM), Armenise/Harvard Laboratory of Integrative Genomics, Pozzuoli, Italy. [11]Department of Advanced Biomedical Sciences, University of Naples "Federico II", Naples, Italy. [12]NEGEDIA S.r.l., Pozzuoli, Italy. [13]Laboratory of Human Lympho-hematopoiesis, INSERM, Paris, France. [14]Department of Medical Biotechnologies, University of Siena, Siena, Italy. [15]Genewerk GmbH, Heidelberg, Germany. [16]Department of Systems Medicine University of Rome Tor Vergata, Rome, Italy. [17]Haematology and Bone Marrow Transplantation Unit, San Raffaele Scientific Institute, Milan, Italy. [18]Department of Translational Medicine, University of Naples "Federico II",

Naples, Italy. ¹⁹School for Advanced Studies, Genomics and Experimental Medicine Program, University of Naples "Federico II", Naples, Italy. ²⁰Department of Pediatric Hematology/Oncology and Cell and Gene Therapy, IRCCS Ospedale Pediatrico Bambino Gesù, Rome, Italy. ²¹Università Cattolica del Sacro Cuore, Rome, Italy. ²²These authors contributed equally: Daniela Cesana, Maria Pia Cicalese. ²³Deceased: Manfred Schmidt. ²⁴These authors jointly supervised this work: Eugenio Montini, Franco Locatelli, Alessandro Aiuti. ✉e-mail: aiuti.alessandro@hsr.it

