## [Peer Review File · Nature Communications]

A case of T-cell acute lymphoblastic leukemia in retroviral gene therapy for ADA-SCIDThis manuscript has been previously reviewed at another journal that is not operating a transparent peer review scheme. This document only contains reviewer comments and rebuttal letters for versions considered at *Nature Communications*.

REVIEWERS' COMMENTS

Reviewer #2 (Remarks to the Author):

The study by Cesana, Cicalese, et. al. investigates the role that LMO2 insertion plays in the first patient to develop T-ALL after receiving retroviral gene therapy (GT) for ADA-SCID. Using Slim PCR method they identified vector integration sites (IS) that included a dominant clone containing an integration ~40 kb upstream of the LMO2 transcription start site. They also identified another IS close to MECOM. They then tracked the clonal proportions post-GT and relative to the T-ALL onset across different lineages to show expansion and selection. Using HiC they were able to identify a possible interaction between the IS and the LMO2 genomic region, providing a possible mechanism of the initial driver of T-ALL in this patient. They then performed an extensive genomic characterization through high coverage (100x) genome sequencing, bulk RNA-seq and single cell RNA-seq and identify somatic genomic structural rearrangements, mutations and changes to transcriptome that are typical of T-ALL. Lastly, they compared results to three other datasets and identified LMO2 as a common insertion site (CIS). The study achieves a very comprehensive characterization of the patient's samples and downstream changes after insertion near LMO2.

The authors have addressed previous comments and concerns in a comprehensive and detailed response. They have now included potential germline variants that could contribute to "patient specific factors" and detailed the reasons why other patients could not be exome sequenced. They have done an analysis of transgene and promoter to determine differences between other therapies that may contribute to observations in ADA-SCID. Lastly, have provided some discussion on the contribution environment and pre-conditioning may have.

Only a few very minor comments remain:

1. Based on the somatic mutations detected and what is known in literature, would it be possible to speculate which are driver mutations and those that are passenger and/or timing of the mutations, particularly the PTEN mutations.
2. Where were the allele frequencies from Supplementary Table 4 taken from? It was not listed in the table key.

Reviewer #4 (Remarks to the Author):

The authors provide the first case of a child with ADA-deficiency who, upon treatment with retrovirus-transduced autologous hematopoietic stem cells, developed T cell acute lymphoblastic leukemia. In very extensive and detailed molecular studies, they link the onset of leukemia to insertional mutagenesis in the LMO2 locus. This is a well-known genomic hotspot predisposing to T-ALL that has previously been identified by the authors and other scientists. The novelty here is the fact several other genomic signatures have been identified which may play a contributing role and thus may explain the uniqueness of this patient. The authors have made use of state-of-the-art technology and have made a huge and laudable effort to explain the peculiarity. The case is of high relevance not only to the scientific but also to the commercial and regulatory world. However, despite honest and remarkable attempts by the authors, the uniqueness cannot be substantiated beyond associative data.

RESPONSE TO REVIEWERS' COMMENTS

Reviewer #2 (Remarks to the Author):

The study by Cesana, Cicalese, et. al. investigates the role that LMO2 insertion plays in the first patient to develop T-ALL after receiving retroviral gene therapy (GT) for ADA-SCID. Using SliM PCR method they identified vector integration sites (IS) that included a dominant clone containing an integration ~40 kb upstream of the LMO2 transcription start site. They also identified another IS close to MECOM. They then tracked the clonal proportions post-GT and relative to the T-ALL onset across different lineages to show expansion and selection. Using HiC they were able to identify a possible interaction between the IS and the LMO2 genomic region, providing a possible mechanism of the initial driver of T-ALL in this patient. They then performed an extensive genomic characterization through high coverage (100x) genome sequencing, bulk RNA-seq and single cell RNA-seq and identify somatic genomic structural rearrangements, mutations and changes to transcriptome that are typical of T-ALL. Lastly, they compared results to three other datasets and identified LMO2 as a common insertion site (CIS). The study achieves a very comprehensive characterization of the patient's samples and downstream changes after insertion near LMO2.

The authors have addressed previous comments and concerns in a comprehensive and detailed response. They have now included potential germline variants that could contribute to "patient specific factors" and detailed the reasons why other patients could not be exome sequenced. They have done an analysis of transgene and promoter to determine differences between other therapies that may contribute to observations in ADA-SCID. Lastly, have provided some discussion on the contribution environment and pre-conditioning may have.

Only a few very minor comments remain:

1. Based on the somatic mutations detected and what is known in literature, would it be possible to speculate which are driver mutations and those that are passenger and/or timing of the mutations, particularly the PTEN mutations.

Immunophenotypic and gene expression signature analyses classified T-ALL into subtypes based largely on differential expression of surface antigen markers and oncogene expression signatures related to stage-specific T-cell developmental arrest (Ferrando et al. 2002, Cancer Cell. 2002;1(1):75-87.; Soulier et al. 2005 Blood 106: 274–286.doi:10.1182/blood-2004-10-3900; Seki et al. 2017 Nat Genet 49: 1274–1281.doi:10.1038/ng.3900). These subtypes are characterized by unique driving aberrations and rearrangements such as MEF2C-activating rearrangements or HOXA gene activation gene fusions in immature/ETP-ALL patients, TLX3/HOXA-activating gene fusion in TLX patients, NKX2-1/2-2 or TLX1 rearrangements in proliferative patients, and TAL1/2, LYL1, or LMO1/2/3 rearrangements in TALLMO patients

(Ferrando et al. 2002; Soulier et al. 2005; Seki et al. 2017). The T-ALL developed in P21 genetic aberrations and immunophenotypic profile that resemble the last group. Thus, based on the literature and our genetic and immunophenotypic data, the vector-driven LMO2 activation and the TAL/STIL recombination event can be considered as the driver mutations that promoted tumorigenesis in our P21 patient. We can speculate that PTEN mutations as well as the other mutations in tumor suppressor genes such as Cdkn2a deletion and the deletion of the long arm of chromosome 6 occur as secondary events in this patient (Kroeze E, *Blood Adv* (2020) 4 (14): 3466–3473, doi.org/10.1182/bloodadvances.2020001822). These events are commonly found in all subtypes of T-ALL and are known as secondary mutations or type B mutations, although PTEN mutations and microdeletions are frequently detected in TAL1- or LMO2-rearranged patients (Liu Y, *Nat Genet.* 2017 August 49(8): 1211–1218. doi:10.1038/ng.3909; Furness CL, *Leukemia* 2018, 32:1984-1993, doi.org/10.1038/s41375-018-0046-8; Mendes R D, *Blood* 24 July 2014, Pages 567-578 <https://doi.org/10.1182/blood-2014-03-562751>).

As stated in the Point by Point reply, we tried to characterize the presence of the T-ALL precursor and potentially the sequential order of the somatic genetic alterations by performing deep sequencing analyses on cfDNA samples collected over time post-GT and before the leukemic outcome. Unfortunately, we failed to detect any of the T-ALL-specific mutations at the different time points analyzed. We also designed 2 ddPCR assays able to detect and quantify the presence of the PTEN-specific mutations identified in compound heterozygosity in the T-ALL. However, also in this case, by analyzing the DNA isolated from T cells and whole blood and whole BM samples collected up to 3 years post HSC gene therapy, we did not identify the presence of such mutations (see below figure for reviewer).

The following sentence was added to the Discussion.

“T-ALL-specific mutations analysed by deep sequencing on cfDNA samples as well as ddPCR specific for PTEN mutations performed on DNA samples collected over time up to 3 years post-GT, failed to detect the presence of any T-ALL mutations. It is possible that the approach was not sensitive enough or that the mutations occurred at later time points (36-57 months post-GT) for which samples were not available”.

2. Where were the allele frequencies from Supplementary Table 4 taken from? It was not

listed in the table key.

The allele frequencies from Supplementary Table 4 were derived from gnomAD_exome, which reports the frequency of each mutation's presence in the human population. We add this information in the Supplementary Table 4.

Reviewer #4 (Remarks to the Author):

The authors provide the first case of a child with ADA-deficiency who, upon treatment with retrovirus-transduced autologous hematopoietic stem cells, developed T cell acute lymphoblastic leukemia. In very extensive and detailed molecular studies, they link the onset of leukemia to insertional mutagenesis in the LMO2 locus. This is a well-known genomic hotspot predisposing to T-ALL that has previously been identified by the authors and other scientists. The novelty here is the fact several other genomic signatures have been identified which may play a contributing role and thus may explain the uniqueness of this patient. The authors have made use of state-of-the-art technology and have made a huge and laudable effort to explain the peculiarity. The case is of high relevance not only to the scientific but also to the commercial and regulatory world. However, despite honest and remarkable attempts by the authors, the uniqueness cannot be substantiated beyond associative data.

We thank you the reviewer for the positive comments about our work.

Regarding the uniqueness of P21, we added a comment in the discussion highlighting that despite the extensive molecular analyses it was not possible to identify the factors that make this patient different and unique from the others. Text changes are highlighted in red.